# LIMIT: LIfestyle and Microbiome InTeraction Early Adiposity Rebound in Children, a Study Protocol

**DOI:** 10.3390/metabo12090809

**Published:** 2022-08-28

**Authors:** Rachele De Giuseppe, Federica Loperfido, Rosa Maria Cerbo, Maria Cristina Monti, Elisa Civardi, Francesca Garofoli, Micol Angelini, Beatrice Maccarini, Eduardo Sommella, Pietro Campiglia, Laura Bertuzzo, Marcello Chieppa, Stefano Ghirardello, Hellas Cena

**Affiliations:** 1Laboratory of Dietetics and Clinical Nutrition, Department of Public Health, Experimental and Forensic Medicine, University of Pavia, Via Bassi 21, 27100 Pavia, Italy; 2Neonatal Unit and Neonatal Intensive Care Unit, Fondazione IRCCS Policlinico San Matteo, 27100 Pavia, Italy; 3Department of Public Health, Experimental and Forensic Medicine—Unit of Biostatistics and Clinical Epidemiology, University of Pavia, 27100 Pavia, Italy; 4Department of Pharmacy, University of Salerno, Fisciano, 84084 Salerno, Italy; 5GlaxoSmithKline (GSK) Consumer Healthcare, Haleon Group, Via Zambeletti s.n.c. 20021 Baranzate, Italy; 6Department of Biological and Environmental Sciences and Technologies (DISTEBA), University of Salento, 73100 Lecce, Italy; 7Clinical Nutrition and Dietetics Service, Unit of Internal Medicine and Endocrinology, ICS Maugeri IRCCS, 27100 Pavia, Italy

**Keywords:** childhood obesity, early adiposity rebound, microbiome, lifestyle, nutrition

## Abstract

Childhood obesity is a strong predictor of adult obesity with health and economic consequences for individuals and society. Adiposity rebound (AR) is a rise in the Body Mass Index occurring between 3 and 7 years. Early adiposity rebound (EAR) occurs at a median age of 2 years and predisposes to a later onset of obesity. Since obesity has been associated with intestinal dysbiosis, we hypothesize that EAR could be related to early microbiome changes due to maternal/lifestyle changes and environmental exposures, which can increase the unhealthy consequences of childhood obesity. LIMIT is a prospective cohort study that aims at identifying the longitudinal interplay between infant gut microbiome, infant/maternal lifestyle, and environmental variables, in children with EAR vs. AR. Methods. The study evaluated 272 mother-infant pairs, enrolled at an Italian neonatal unit, at different time points (T0, at delivery; T1, 1 month; T2, 6 months; T3, 12 months; T4, 24 months; T5, 36 months after birth). The variables that were collected include maternal/infant anthropometric measurements, lifestyle habits, maternal environmental endocrine disruptor exposure, as well as infant AR. The LIMIT results will provide the basis for early identification of those maternal and infant modifiable factors on which to act for an effective and personalized prevention of childhood obesity.

## 1. Introduction

Obesity is a worldwide epidemic and is one of the most critical public health challenges. The number of people with obesity has tripled in Europe compared to 40 years ago [1]. In the certain EU Member States, obesity prevalence in children is reaching well over 30% with a significant impact on their health, affecting physical and psychological abilities, which can further worsen health costs and quality of life in adulthood [1,2]. Indeed, obesity is already responsible for 2–8% of the health costs and 10–13% of the deaths in different parts of Europe [1].

Given that childhood obesity is a strong predictor of obesity in adulthood [3,4] and that its prevalence is projected to increase further by 2030, obesity poses a major threat to the future increase of all non-communicable diseases (NCDs) including diabetes, cardiovascular disease, hypertension, stroke, and some cancers, leading to an increased risk of premature mortality [5,6]. 

Since 1980, the global prevalence of childhood obesity has increased by 47% [7]. In 2016, 40 million children under 5 years of age and more than 330 million children and adolescents between 5 and 19 years old were estimated to be affected by overweight or obesity [8]. Although recently, obesity prevalence among 7–8-year-old children has shown a slight decrease in several EU countries (e.g., Greece, Italy, Portugal, Slovenia), almost 1 in 8 children have excessive weight gain (14% of males and 10% of females aged 7 to 8 in 23 EU countries) [1] with critical health-related consequences, including depression, behavioral disorders, stigma, and poor self-esteem [9,10].

This global increase in obesity is driven by several lifestyles and environmental factors [11] (e.g., family environment related to a sedentary lifestyle, unhealthy dietary habits and eating behavior, poor quality of sleep, increased screen time), which expose them to biological, psychological, and social threats [12].

Finding an early predictor of later excessive body weight can help early interventions to counteract the obesity epidemic. A promising predictive index for the early onset of obesity is early adiposity rebound (EAR) [13,14]. In the physiological growth process, a rapid increase in the Body Mass Index (BMI) occurs during the first year of life; then, the BMI decreases and reaches the minimum value around 6 years of age, before starting a sustained increase [15]. Evidence shows that when adiposity rebound (AR) occurs earlier, at a median age of 3 years old [16], it is associated with an increased risk of overweight/obesity and central body fat depot [16]. Indeed, in a study conducted on 76 adolescents affected by obesity, Péneau et al. [17] showed that 97% of them experienced EAR at a median age of 2 years old, confirmed by others [3,18,19], who described a strong relationship between AR timing and the subsequent risk of developing obesity, as well as impaired glucose tolerance and type 2 diabetes, metabolic syndrome, coronary heart disease, and polycystic ovary syndrome [3,20,21].

Those remarkable observations suggest that obesity in adolescence and adulthood is somehow “programmed” much earlier in life, possibly even during periconception [22]. Recent evidence has explained the impact of prenatal and early postnatal events in promoting obesity risk later in life [23]. Mainly, embryonic and fetal ages are vulnerable to environmental insults, and acquired variations may endure through generations, despite the absence of continuous exposure. This critical period for offspring’s future health corresponds to “the first 1000 days’’, which runs from conception and continues up to two years of life [24].

In this same period, the intestinal microbiota changes shape and conforms to what will later be the definitive microbiota in adulthood [25]. Cumulative evidence suggests that the gut microbiome is an essential mediator between different factors such as genetics, diet, exercise, environmental substances exposure (e.g., exposure to endocrine-disrupting chemicals (EDCs)) and the pathophysiology of obesity and obesity-related metabolic disorders, driving NCDs’ intergenerational nature [25,26,27]. Gut microbiota shaping and role start during the fetal period with microbes exchanged from the mother to the fetus [28,29,30], suggesting that this ecosystem composition, especially during “the first 1000 days” window, is influenced by several maternal and infant factors (e.g., maternal adherence to healthy dietary patterns, maternal physical activity, type of birth, maternal use of medications, type of breastfeeding and weaning, sleep pattern, and infant dietary habits) [25,31,32,33,34,35]. Our previous study (Alimentazione Mamma e bambino nei primi Mille giorni (A.MA.MI)) demonstrated that delivery mode, maternal pre-pregnancy BMI, and type of feeding influenced the infant microbiota composition [36,37]. Besides, women enrolled in the A.MA.MI project reported widespread exposure to products containing EDCs (e.g., monoethyl phthalate (MEP) and bisphenol A (BPA)) [38] which are reported to be associated with early childhood obesity patterns, increasing the risk of developing obesity and NCDs during life [27], as they can cross the placenta and concentrate in the circulation of the fetus or can be transferred from mother to baby through breast milk [39,40].

The period from conception up to 2 years of age is the most critical for the induction of pathophysiological derangements, which may affect the health status of the offspring in the short and long term [41,42], leading to the development of different diseases including inflammatory bowel diseases (IBDs) and other autoimmune diseases [43,44], allergy [45], obesity [46], and many NCDs [25,31,32,33,34,35,46,47,48,49].

Although the AR phenomenon appears well documented, the mechanisms underlying an EAR have yet to be elucidated. Based on the above-mentioned considerations, the LIfestyle and Microbiome InTeraction early adiposity rebound in children (LIMIT) study aims at identifying the differences in the longitudinal interplay among the infant intestinal microbiome, the infant/maternal lifestyle, and environmental exposure (e.g., EDCs), in children with EAR vs. AR. The acquired knowledge may allow the definition of an AR-healthy phenotype to design a model able to predict the risk of developing EAR in children. Thus, a prognostic index like EAR could be linked to early dysbiosis, which could be corrected by personalized lifestyle interventions to prevent childhood obesity development and stop this epidemic.

## 2. Material and Methods

LIMIT is a prospective cohort study aimed at:Evaluating whether the (i) intra-individual abundance (alpha-diversity) and the (ii) inter-individual abundance (beta-diversity) of the intestinal microbiota in the first 6 months of life is associated with EAR within 36 months of life (primary outcome);Investigating the correlation among the (i) maternal factors (pre-pregnancy BMI, weight gain during pregnancy, dietary habits, and physical activity), (ii) infant factors (delivery mode, feeding and weaning mode, dietary habits, sleeping habits), and (iii) environmental exposure factors (exposure to maternal endocrine-disrupting chemicals) and infant intestinal microbiome composition (alpha- and beta-diversity) at delivery and different time points of assessment.

### 2.1. Participants

To address LIMIT’s purposes, 272 mother-infant dyads were consecutively enrolled during the pre-hospital care before birth at the UOC Neonatology and Neonatal Intensive Care, Fondazione IRCCS Policlinico San Matteo of Pavia, and then, evaluated at a different time point of assessment: at delivery (T0); 20–30 days post-delivery (T1); 6 months post-delivery (T2); 12 months post-delivery (T3); 24 months post-delivery (T4); and 36 months post-delivery (T5). Participants were included in the study according to the following inclusion criteria: healthy term (gestational age between the 37th week and the 42nd week) newborns; parents’ ability to speak and understand the Italian language; parents’ ability to sign the informed consent and to fill in the questionnaires; elective caesarean section and vaginal delivery (1:1 ratio).

The exclusion criteria were: infants with genetic/congenital diseases; infants requiring intensive care immediately after birth; infants born with insulin-dependent diabetes-mellitus; infants with severe intrauterine growth retardation (below the 3rd percentile) and weighing <10th percentile, according to the Italian Neonatal Study (INeS) charts [50]; infants who required resuscitation; infants with meconium-tinged fluid; infants receiving antibiotic therapy and/or maternal fever >38 °C in labor; maternal hyperthyroidism during pregnancy; the presence of gestational diabetes; infants born from parents refusing to sign the informed consent. 

The study followed the Declaration of Helsinki. Written informed consent was obtained from the parent or legal guardians.

Ethical approval was granted by the Ethical Committee of IRCCS Policlinico San Matteo (Pavia) (protocol number: 0020200/22; Accepted: 11 April 2022). The LIMIT protocol has been registered on clinicaltrial.gov (accessed on 23 August 2022) (NCT04960670). 

At the different time points of assessment, infant auxological parameters (length/height, weight, and head circumference) and maternal anthropometric measurements (height, weight, and waist circumference) were measured.

Information about the mothers’ pathological/physiological history, pregnancy (type of current pregnancy, spontaneous or assisted pregnancy, number of previous pregnancies); socio-demographic data (e.g., socio-economic level of the family); pre-gestational/gestational anthropometrics parameters (e.g., maternal weight status before pregnancy, weight gain during pregnancy); infant medical history; maternal and infant use of antibiotics, probiotics, and supplements; maternal lifestyle factors (e.g., dietary habits, physical activity level, smoking habits before, during, and after pregnancy, feeding attitude), and infant ones (feeding and weaning mode, dietary habits, sleeping habits, physical activity level), as well as delivery mode, family environment (e.g., how many people live in the house, infant’s exposure to passive smoking) and maternal environmental exposures (e.g., EDCs), by means of interview, previously validated questionnaires or adapted questionnaire, by using a forwarding–back translation (FBT) [51] were collected (Table 1).

Biological samples, including infant stool samples and maternal urine samples at different time points, were collected to assess the infant gut microbiota composition and maternal EDC levels (Table 2).

#### 2.1.1. Infant Auxological Parameters

Length (cm) or height (cm) (T5), weight (Kg) (T0–T5), and head circumference (T0–T4) were assessed at pre-determined time points (Table 1). 

Briefly, the body weight of unclothed children was measured using a balanced weight scale (accuracy ± 100 g), following standardized procedures [52]. The length was measured using an infantometer; in the case of the impossibility of stretching both legs straight in the correct position, the examiner would ensure that at least one leg was straight with the foot flexed against the footrest. 

Standing height was measured using a Harpenden stadiometer with a fixed vertical backboard and an adjustable headpiece. The measurement was taken of the child in the upright position, without shoes, hands at the sides, aligning the head with the Frankfort horizontal plane. The child was instructed to stand as tall as possible, take a deep breath, and hold this position to capture the result [53]. Head circumference was measured according to the standard evaluation methods [52].

#### 2.1.2. Infant Adiposity Rebound Monitoring

In the LIMIT project, auxological parameters (weight, length/height) and the BMI Z-score (BMI-z) were, respectively, measured and calculated by using the World Health Organization’s age and sex-specific standards [60]. In this study, the timings of AR were set as a binary variable, and the children were categorized into early or non-early AR. EAR was defined as an increase of more than two percentile thresholds in the BMI curve between 1 and 36 months of age (maximum age of the children in the study). As previously reported by Pereira-da-Silva and Virella [61], the inaccurate measurement of the length, if squared (BMI), magnifies the error of the index in which it is included, while losing the ability to differentiate overestimation from underestimation. In the LIMIT study, the EAR was evaluated by two investigators: one trained and one under training, in a blinded fashion, using BMI z-scores. For this reason, the inter- and intra-observer variability were previously evaluated according to the standardized protocols described by Stomfai and colleagues [62].

#### 2.1.3. Maternal Anthropometric Parameters

Anthropometric measures of each woman were collected as follows: pre-pregnancy weight (Kg) was self-reported, while height (cm) was measured on all the subjects enrolled without shoes using a stadiometer (accuracy ± 1 mm). Pre-pregnancy Body Mass Index (BMI, kg/m^2^) (T0) was then calculated as the ratio of the weight, expressed in kg, to height in meters squared [52]. The pre-pregnancy BMI calculation allows defining and comparing mothers with pre-pregnancy normal-weight mothers (BMI < 25 kg/m^2^) vs. mothers with overweight/obesity (BMI ≥ 25 kg/m^2^), as also previously described in the A.MA.MI project [36].

Similarly, weight gain during gestation was self-reported and used to define women with excessive gestational weight gain (EGW) vs. women with optimal gestational weight gain (GWG), according to the Institute of Medicine (U.S.) and National Research Council (U.S.) Committee to Reexamine IOM Pregnancy Weight Guidelines [63], as well as previously described in the A.MA.MI project [36]. In particular, the recommended weight gains for women with underweight (BMI < 18.5 kg/m^2^), normal weight (BMI = 18.5–24.9 kg/m^2^), overweight (BMI = 25.0–29.9 kg/m^2^), and obesity (BMI > 30.0 kg/m^2^) are 12.5–18, 11.5–16, 7.0–11.5, and 5.0–9.0 kg, respectively [63]. After delivery, body weight (Kg) was measured (T1–T5) with mothers in underwear, on a balanced weight scale (accuracy ± 100 g). BMI (kg/m^2^) was then calculated as reported above. Waist circumference (WC, cm) (T3–T5) was measured to the nearest centimeter with a flexible steel tape measure with the participants standing, with crossed arms resting on opposite shoulders, after a slight exhalation. It was measured on the horizontal plane between the lowest portion of the rib cage and the uppermost lateral margin of the right ileum [52]. The waist/height ratio (T3–T5) and abdominal adiposity index (cut-off < 0.5) [52] were also calculated.

#### 2.1.4. Maternal Dietary Habits 

Food consumption frequency (FF) and dietary habits (DH) at T0, T1, and T3 were investigated using a previously validated self-administered dietary questionnaire [62]. Before administration, two out of the nine sections of the original questionnaire, developed and validated for a teen Italian population [55], were adapted by two dieticians to the adult population, as previously described elsewhere [38].

In brief, the FF section (18 items) investigates the daily consumption of typical foods and beverages, such as bread, pasta, cereal-based products, fruit and vegetables, milk and yoghurt, tea and coffee, and weekly consumption of other foods such as meat and meat products, fish, eggs, cheese, legumes, sweets, and alcohol [55]. Each section consists of a multiple-choice questionnaire, which includes the following answer categories: always, often, sometimes, never [55]. The score assigned to each answer ranges from 0 to 3, with the highest score given to the healthiest one and the lowest score to the least healthy one [55]. The DH section (14 questions) is designed to investigate dietary habits including breakfast consumption, the daily number of meals, fruit and vegetable intake, and consumption of non-alcoholic or alcoholic beverages. In this section, some questions aim to assess whether the number of servings consumed meets the recommendations [55]. Eight questions have the following response categories: always, often, sometimes, never; the score assigned to each answer ranged from 0 to 3 [55]. The other six questions have four differently structured response categories, and the score ranges from 1 to 4. In both cases, the maximum score is assigned to the healthiest one and the minimum score to the least healthy [55]. The total score is then divided into tertiles, where the lowest one corresponds to “inadequate eating habits”, the average one corresponds to “partially satisfactory eating habits”, and the highest one corresponds to “satisfactory eating habits”, according to the Italian National Dietary Guidelines [64]. 

#### 2.1.5. Mediterranean Dietary Pattern Adherence 

Mediterranean dietary (MD) adherence was assessed at T0, T1, T4, and T5 using the MEDI-LITE score obtained from a previously validated questionnaire [57]. The questionnaire investigates the frequency of consumption of nine classes of food: (i) fruit, (ii) vegetables, (iii) cereal grains, (iv) legumes, (v) fish and fish products, (vi) meat and meat products, (vii) dairy products, (viii) alcohol intake, and (ix) olive oil [57]. The score obtained from the questionnaire ranges from 0 to 18, where the highest value corresponds to the highest MD adherence [57]. In brief, for fruit, vegetables, cereal grains, legumes, and fish, a score of 2 was assigned for high-frequency consumption, a score of 1 for moderate frequency consumption, and a score of 0 for low-frequency consumption [57]. On the contrary, for meat/meat products and dairy products, low-frequency consumption scores 2, moderate frequency consumption scores 1, and high-frequency consumption scores 0 [57]. For alcohol, the categories related to the alcohol unit (1 alcohol unit = 12 g of alcohol) were used by giving two points to the middle category (1–2 alcohol units/d), 1 point to the lowest category (1 alcohol unit/d), and 0 points to the highest category of consumption [57]. Last, women who report regular use of olive oil score 2; women who report frequent use of olive oil score 1; those who consume it occasionally score 0 [57]. The final score varies from 0 (low adherence) to 18 (high adherence); overall, women reporting a score at or higher than 9 will have a significantly increased possibility of being adherent to the MD, as previously reported [57]. 

#### 2.1.6. Maternal Smoking Habits

Smoking habits were investigated considering women who never smoked, quit smoking before pregnancy, or during pregnancy, smoked during pregnancy or resumed smoking after childbirth, or after lactation. The packages of cigarettes/per year were recorded.

#### 2.1.7. Maternal Physical Activity Level

Maternal physical activity was assessed by the short form (7 items) of a validated country-/language-specific questionnaire (International Physical Activity Questionnaire (IPAQ—SF)) at T0 [58] (downloadable from https://sites.google.com/site/theipaq/questionnaire_links accessed on 3 May 2022). The questionnaire provides an estimate of the metabolic equivalent of task (MET-min) per week, which is calculated as follows: METs = MET level x × minutes of activityx × events per week. Physical activity level is classified according to METs into sedentary (total METs < 699), moderate (total METs between 700 and 2519), and high (total METs > 2520) [58].

#### 2.1.8. Maternal Feeding Attitude

The Iowa Infant Feeding Attitude Scale (IIFAS) was used to assess the maternal attitude towards infant feeding methods [54] and to predict breastfeeding intention and exclusivity. In brief, the IIFAS is a standardized interview questionnaire on participants’ characteristics and habits (age, residence, education, self-assessed socioeconomic status, relationship status, work before pregnancy, back to work after maternity leave, mode of delivery, parity, way of feeding previous babies, planned way of feeding this newborn). The scale consists of 17 items with a 5-point Likert scoring from 1 (strongly disagree) to 5 (strongly agree). Items 1, 2, 4, 6, 8, 10, 11, 14, and 17 are reverse-scored. 

The total score ranges from 17 to 85 with a higher score reflecting a positive attitude toward breastfeeding [54]. In brief, scores ranging between 70 and 85 correspond to a positive attitude toward breastfeeding; scores ranging between 49 and 69 correspond to a neutral attitude; scores ranging between 17 and 48 correspond to a positive attitude towards formula feeding [54].

#### 2.1.9. Infant Dietary Habits

The validated INTERGROWTH-21st Food Frequency Questionnaire [56] was used to collect infants’ dietary habits at 1 year of age (T3) [56]. In brief, the questionnaire investigates (i) infants’ feeding upon discharge through the first year of life, (e.g., During the 1st year of life, have you given your child expressed milk? How old was your child when you stopped exclusively breastfeeding? Is your child following any special diet?); (ii) food frequencies over the past 28 days (e.g., Did your son eat cooked cereal? Did your son eat legumes? Did your son drink soft drinks?). Each question has the following responses: 1–3 times/month, 1–3 times or >3 times/week, 1–3 times or >3 times/day, and not applicable [56].

However, the INTERGROWTH-21st Food Frequency Questionnaire [56] does not take into consideration the protein content of infant formula, which is reported to be among the main causes influencing early excessive weight gain, including in Mediterranean European countries [65,66]. Thus, this aspect was further and separately asked.

#### 2.1.10. Infant Sleeping Habits

The Infant Sleep Questionnaire (ISQ) was used to assess infant sleep (T3–T5) by assessing sleep quality among infants in the last month of life [59]. The questionnaire consists of 8 multiple-choice items related to the length, quality, and mode of sleep. The ISQ can classify infant sleep problems in three ways: (i) Richman’s criteria (i.e., sleep problem defined as settling or waking problem occurring 5 or more nights per week, plus one or more of the following: taking more than 30 min to settle, waking 3 or more times per night, awake for more than 20 min during the night, sleeping in parents’ bed because upset) [67]; (ii) maternal criteria (e.g., Do you think that your baby has sleeping difficulties?) using a 4-point scale (no problem, mild problem, moderate problem, or severe problem) [59]; and (iii) via a severity score [59].

Overall, the questionnaire returns a score ranging from 0 to 38, as previously reported [59].

For the Richman criteria, the ISQ cut-off of 12 or above appeared to detect more severe sleep problems; for the maternal criteria, the ISQ cut-off of 6 or above appeared to detect a wider range of sleep problems [59].

#### 2.1.11. BPA and Phthalates Daily Exposure

The determinants of exposure to BPA, phthalates and their secondary metabolites (e.g., use of plastic products, presence of synthetic material at the gym or during recreational activities, presence of PVC in the home and working environments, consumption of packaged food in plastic containers or tetra-packed food, consumption of pre-cooked food, disposable plastic use, regular use of plastic utensils for cooking) were recorded at T1 and T3, as previously described by our group [38]. Several questions were extrapolated from the questionnaire designed by the “LIFE PERSUADED” project (“Phthalates and bisphenol A biomonitoring in Italian mother-child pairs: the link between exposure and juvenile diseases”) [38]. The questionnaire evaluates the association between demographic and lifestyle variables potentially related to exposure to DEHP/BPA and secondary metabolites in children and their mothers [38]. Data are expressed as the frequency of lifestyle habits over-mentioned and listed in percentage.

#### 2.1.12. Infant Stool Sample Collection and Microbiome Analysis 

For each infant, stool samples were collected to analyze the intestinal microbiota. The first stool sample (T0) was collected by a member of the UOC of Neonatology and Neonatal Intensive Care, whereas subsequent samples (T1–T5) were collected by the infants’ parents after they are accurately instructed by a member of the research team on how to collect, store, and transport infant stool samples, according to the following inclusion criteria: (i) absence of fever and gastroenteritis; (ii) absence of diarrhea for more than 24 h in the previous 7 days; (iii) no antibiotic treatment of the child in the previous 7 days.

The collected samples were then anonymized by the researchers who received them, using a progressive code and stored at −80 °C, until the analysis. 

Subsequently, the fecal samples were shipped on dry ice and whole metagenomic shotgun sequencing [68] was applied.

Metagenomic libraries were generated with a Nextera XT DNA Sample Prep Kit (Illumina, San Diego, CA, USA), and sequencing was carried out on the HiSeq2500 platform (Illumina) at a targeted depth of 5.0 Gb (100 bp paired-end reads). Shotgun metagenomics sequencing samples were pre-processed as previously described by others [69].

#### 2.1.13. Maternal Urinary Sample Collection and Endocrine-Disrupting Chemicals’ Analysis 

Maternal urinary samples were collected at T1 and at T3 to measure the BPA levels; metabolite of diethyl phthalate (DEP), named monoethyl phthalate (MEP); mono isobutyl-phthalate (MIbP) as a metabolite of -n-butyl phthalate (DnBP); metabolites of DEHP, such as mono (2-ethyl-5-hydroxylhexyl) phthalate (MEHHP) and mono (2-ethylhexyl) phthalate (MEHP); mono benzyl phthalate (MBzP) and metabolite of benzyl butyl phthalate (BBP) (Table 2). In addition, creatinine concentrations were also measured. 

In particular, mid-stream clean urine samples instead of first or 24 h urinary samples, according to Lee et al. [70], were collected into polypropylene cups for urine culture, and then, mothers were asked to store the cups in a refrigerated place until transportation. The urine samples were then transported under refrigeration, aliquoted into phthalate-free tubes, and stored at –80 °C at the Neonatal Immunology Laboratory of the UOC Neonatology and Neonatal Intensive Care, until the analysis.

BPA and phthalates levels were measured by UHPLC-MS/MS (Shimadzu, Milan, Italy). 

The UHPLC system consists of two LC 30 AD pumps, a SIL 30 AC autosampler, a CTO 20 AC column oven, and a CBM 20 A controller, and the system is coupled online to a triple quadrupole LCMS 8050 (Shimadzu, Kyoto, Japan) equipped with an electrospray ionization (ESI) source. Although EDCs’ threshold levels have not been defined universally, cut-offs derived by human biomonitoring (HBM) in urine have been suggested as reference values for humans [71].

Creatinine levels were measured by routine methods on an Atellica^®^ CH Analyzer (Siemens Healthcare Diagnostics Inc., Tarrytown, NY, USA) and expressed as mol/L.

### 2.2. Statistical Analyses

Preliminary results of the “Alimentazione Mamma e bambino nei primi Mille giorni”— “Mother-infant nursing in the first thousand days” (A.MA.MI) project [36] on 63 mother-infant pairs were used to hypothesize which distribution of alpha-diversity (Shannon index) [72] we expect in infants. In particular, in the A.MA.MI project, at six months post-delivery, the Shannon index in infants was normally distributed, with a mean of 2 and a standard deviation of approximately 0.5 [37]. 

The sample size of the LIMIT project is calculated by fixing a priori the power (90%) and the level of statistical significance (alpha = 0.01) and estimating that the EAR (primary endpoint of the study) can occur in 33% of children. Assuming an average difference in the alphadiversity at T2 (six months after delivery) between EAR and AR infants of 0.25 units, corresponding to an average effect on the risk to develop EAR (Cohen’s D of 0.5), the sample size needed was about 272 mother pairs. A variation in the microbial community was considered an indicator of dysbiosis and was tested for association with the EAR.

The collected sample was described using the appropriate summary measures: means and standard deviations or median and interquartile range (IQR) for the quantitative variables, frequency distributions for the qualitative variables in the overall global sample in the EAR or AR children groups.

To compare the alpha-diversity (Shannon index) [72] between EAR and AR, Student’s *t*-test or the Mann–Whitney U-test is used, depending on whether the data are normally distributed. The proportion or presence/absence of specific microbial units in EAR children vs. those with AR was also compared, with adjustments for multiple comparisons.

Furthermore, a logistic model was used to evaluate the association between the presence of EAR at 36 months and the variation (T2–T0) in the alpha-diversity (Shannon index) [72], adjusting for other factors (maternal weight status before pregnancy, weight gain during pregnancy, weight during pregnancy, gestational exposure, maternal exposure to environmental pollutants, perinatal and postnatal exposure). The adjusted odds ratios (ORs) and 95% confidence intervals were used as measures of effect and precision, respectively.

Student’s *t*-test or the Mann–Whitney U-test, as appropriate, were used to compare the variation (T2–T0) in the beta-diversity (Sörenses index) [73] between EAR and AR at 36 months. A logistic model was used to evaluate the association between the presence of EAR at 36 months and the variation of beta diversity, adjusting for all other factors described above.

Pearson or Spearman correlation, principal coordinate and component analyses (PCA and PCoA), and cluster analysis were used to evaluate changes in gut microbiota composition over the various time points of assessment.

Finally, generalized linear models for repeated measurements were used to evaluate the factors influencing the variation of newborn gut microbiota alpha- and beta-diversity, during the whole period (the first 36 months of life), considering robust standard errors in the intra-group correlation.

## 3. Discussion

Obesity in childhood is associated with a wide range of serious health complications and an increased risk of premature NCDs, including diabetes and heart disease [5,6]. The rising prevalence of obesity worldwide represents a growing burden of NCDs and a significant public health issue due to increasing public service health costs [74,75]. Several factors, including environmental, dietary, lifestyle, host, and genetic ones, have been attributed to the development of childhood obesity; however, none of them completely elucidate the increase in the prevalence of childhood obesity, which is preceded by EAR [16]. Rolland-Cachera et al. [13] appraised that adolescents who have experienced EAR (<5.5 years) were more frequently affected by obesity than those who had been rebounders later (>7 years). Thus, between 7% and 30% of 4.2 million babies born in 2019 in the EU, equivalent to 294,000–1.26 million, are expected to experience EAR. 

Evidence shows a significant role in gene-environment interactions where one’s genetic profile influences the ability to deal with the obesogenic impact of some environmental factors [76]. Alteration in the composition of human-associated microbial communities and metabolite profiles related to the gut microbiota can offer deep insights into the role of the microbiome in the development of adiposity and obesity, as well as the exposure to lifestyle, dietary, and other environmental factors [77].

Our protocol, named LIfestyle and Microbiome InTeraction (LIMIT), describes the rationale and design of a prospective, longitudinal, observational study (LIfestyle and Microbiome InTeraction) aiming at investigating the composition and development of the intestinal microbiota in infants from 0 to 36 months, according to the infant growth trajectories, the maternal and infant lifestyle, and the exposure to environmental factors. Several studies either explored the interplay between infant adiposity rebound and obesity [13,15,78] or investigated maternal and infant factors shaping the infant gut microbiota composition [79,80,81]. To the best of our knowledge, our protocol is the first that has the ambition to identify the longitudinal interplay between infant intestinal microbiome, maternal and infant lifestyle factors, as well as EDCs signatures in children showing EAR and in those showing regular AR.

Indeed, LIMIT was designed to consider the following assumptions: (i) during the perinatal period, several maternal factors such as health status, socioeconomic status, microbiome composition, dietary habits and other lifestyle factors, pre-pregnancy weight and weight gain during gestation, use of antibiotics and delivery mode may influence the early stages of microbial colonization in the infant [31,82,83], besides that most of those maternal factors have been shown to be positively associated with EAR in young children [14,84]; (ii) during the postnatal period, several infant factors (e.g., energy intake, type of feeding, sleep patterns, speed of growth) have been shown to be associated with EAR [85,86,87]; (iii) obesity may be programmed in utero; it has been shown that the gut microbiota can be transmitted, at least partially, from mother to child via maternal contact, as well as by environmental contact [85], and epigenetics has also been shown to play a key role in transmitting obesity risk to infants [87]; (iv) dysbiosis (imbalance in the gut microbiota) has been linked to the development of obesity by multiple mechanisms. The gut microbiota is involved in molecular crosstalk with the host, affecting digestive tract physiology and polysaccharides digestion, which directly increases the energy harvest of the host physiology, metabolism, and inflammatory status, promoting adipogenesis and causing weight gain [88,89,90,91]; (v) EDCs are a class of obesogenic substances that are persistent in the environment and humans [92] and are also involved in the gut microbiota’s modulation [91].

Compared to other studies, to the best of our knowledge, this is the first study to simultaneously analyze the association between prenatal, perinatal, and postnatal factors, infant microbiota composition, and age at AR. Indeed, the strength of the LIMIT study is evaluating AR and child growth patterns rather than BMI in a specific early childhood period, considering the composition of the child’s gut microbiota as a potential driver towards increased risk of developing EAR. It is noteworthy that a high initial BMI followed by a late AR may be followed by a subsequent normal or low body weight, while a low initial BMI with an EAR may be associated with excessive weight gain and the development of metabolic disease in a child’s future life [76,93]. Besides, acquiring more information on the intestinal microbiota’s complexity, the host’s interaction, and the exposure factors during the first 36 months of life, LIMIT allows dissecting the mechanisms that lead to the EAR and potentially building a predictive model for the early diagnosis of obesity. 

Despite the novelty of the research question, the study also reports some limitations as it did not take into account the genetic susceptibility to obesity, which has been demonstrated to play a predictive role in the timing of age at AR, as recently reported by Cissè and colleagues [53].

Again, another important strength is that LIMIT analyzes the mother-infant dyad with a holistic approach rather than a reductionist one to (i) figure out the biological pathways during the perinatal period and the first 36 months of life that trigger obesity later in life; (ii) uncover the interplay between maternal-infant factors, environmental exposures, and infant gut microbiota composition to underpin the protective role vs. the unfavorable role of AR timing; (iii) provide evidence for a personalized approach to interventions to fight obesity before and during fetal programming. Therefore, to make the most of this approach, the LIMIT project uses previously validated questionnaires for the collection of maternal and infant lifestyle habits data and standardized protocols for the analysis of the infant gut microbiota composition and the maternal urinary levels of EDCs.

However, even in this case, there are some limitations to consider. First, some factors, including the protein content of infant formula, which is reported to be among the main causes influencing early excessive weight gain [65,66], are not evaluated by the tool we chose for assessing the infants’ dietary habits. For this reason, this aspect will be investigated with separate questions that are not part of the validated questionnaire. Similarly, as far as we know, since a validated questionnaire on EDC exposure assessment is not available, the LIMIT project uses questions that were extrapolated from a large cohort study (e.g., the LIFE PERSUADED study) [38]. Last, since EDCs and the gastrointestinal microbiota interact via multiple mechanisms [94,95], the present research investigated only the association between the maternal BPA and phthalates levels and the infant gut microbiota composition, without taking into consideration how this exposure impacts the maternal hormonal profile.

Overall, LIMIT aspires to act very early in children’ s life to lead to a good foundation of health and the prevention of obesity, which needs an urgent public health resolution. Indeed, health care worldwide is facing the challenge of a rising cost burden due to the increase in obesity and NCDs. Substantial public awareness campaigns/policies for promoting healthy lifestyles, minimizing unhealthy environmental impact, and addressing health inequities to fight obesity should involve policymakers, private sector partners, medical professionals, and the public at large. Besides the “one size fit all” approach conducted till now will unlikely succeed due to the multifactorial nature of obesity; instead, a personalized approach is recommended. LIMIT proposes a personalized life-course approach during early life development, acknowledging that influences operating during “the first 1000 days” window can lay the foundation for health and wellbeing throughout the life course.

## Figures and Tables

**Table 1 metabolites-12-00809-t001:** Variables and time of collection.

Variables	Methods		Time Point of Assessment					
			T0	T1	T2	T3	T4	T5
Anthropometric parameters	**M**	Weight status before/during pregnancy (weight, height, BMI); direct measurement [52]	✓	✓	✓	✓	✓	✓
	**B**	Weight, length, and head circumference; direct measurement [52,53]	✓	✓	✓	✓	✓	✓
Medical history	**M**	Clinical record	✓	✗	✗	✗	✗	✗
	**B**	Clinical record	✓	✓	✓	✓	✓	✓
Antibiotics/probiotics/ supplement use	**M**	Interview	✓	✓	✓	✗	✗	✗
	**B**	Clinical record and interview	✗	✓	✓	✓	✓	✓
Feeding/weaning	**B**	Interview	✗	✗	✓	✗	✓	✗
Feeding	**M**	Iowa Infant Feeding Attitude Scale (IIFAS) (adapted by forwarding–back translation) [54]	✗	✓	✗	✗	✗	✗
Dietary habits	**M**	Previously validated questionnaire [55]	✓	✓	✗	✓	✗	✗
	**B**	INTERGROWTH [56]	✗	✗	✗	✓	✓	✓
Adherence to Mediterranean dietary pattern	**M**	MEDI LITE score [57]	✓	✓	✗	✗	✓	✓
Physical activity level	**M**	International Physical Activity Questionnaire (IPAQ-SF; short form) [58]	✓	✗	✗	✗	✗	✗
	**B**	Interview	✗	✓	✓	✓	✓	✓
Smoking habits	**M**	Interview (before/during/after pregnancy)	✓	✓	✓	✓	✓	✓
Sleep	**B**	The infant sleep questionnaire (ISQ) (adapted by forwarding–back translation) [59]	✗	✗	✗	✓	✓	✓
EDC levels/exposure	**M**	Questionnaire [38]	✗	✓	✗	✓	✗	✗
Socio-economic status (SES)	**M**	Interview	✓	✗	✗	✗	✓	✓
Family environment	**M**	Interview	✓	✓	✓	✓	✓	✓

M: mother; B: baby; T0: time of delivery; T1: 20–30 days after delivery; T2: 6 months after delivery T3: 12 months after delivery; T4: 24 months after delivery; T5: 36 months after delivery; BMI: Body Mass Index; EDCs: endocrine-disrupting chemicals.

**Table 2 metabolites-12-00809-t002:** Endocrine-disrupting chemicals measured in maternal urinary samples.

Phthalate	Secondary Metabolite	Time Point of Assessment					
		T0	T1	T2	T3	T4	T5
Bisphenol A (BPA) (μg/L)	Monoethyl phthalate (MEP) (μg/L)	✗	✓	✗	✓	✗	✗
Diethyl phthalate (DEP) (μg/L)		✗	✓	✗	✓	✗	✗
Butylbenzyl phthalate (BBzP) (μg/L)	Monobenzyl phthalate (MBzP) (μg/L)	✗	✓	✗	✓	✗	✗
Diisobutyl phthalate (DiBP) (μg/L)	Monoisobutyl phthalate (MiBP) (μg/L)	✗	✓	✗	✓	✗	✗
Di-(2-ethyl-hexyl) phthalate (DEHP) (μg/L)	Mono- (2-ethyl-hexyl) phthalate (MEHP) (μg/L) Mono- (2-ethyl-5-hydroxy-hexyl) phthalate (5OH-MEHP) (μg/L) 5oxo-MEHP Mono- (2-ethyl-5-oxo-hexyl) phthalate (5oxo-MEHP) (μg/L)	✗	✓	✗	✓	✗	✗

T0: time of delivery; T1: 20–30 days after delivery; T2: 6 months after delivery T3: 12 months after delivery; T4: 24 months after delivery; T5: 36 months after delivery.

## Data Availability

The data presented in this study are available on request from the corresponding author. The authors guarantee that the data that will be shared will comply with the consent given by the participants on the use of confidential data.

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
