# Peer review of "LIMIT: LIfestyle and Microbiome InTeraction Early Adiposity Rebound in Children, a Study Protocol"

_metabolites, 2022, doi:10.3390/metabo12090809_

Round 1
Reviewer 1 Report
Overall, it is an interesting study. However, as a protocol article, missing some design rationales and detailed descriptions.
Method
No power calculation, why choose 272 as the sample size?
For exclusion criteria, excluding SGA, Are you including LGA? Why or why not?
clinicaltrial.gov protocol stated the sample collection is 0-48months, why the current protocol is up to 36 months?
Do BPA and Phthalates Daily Exposure seem to be an important part of the trial? but how this is related to early adiposity rebound is not clearly stated.
Participants: what is your recruitment method? By poster or clinician refer?
What is your microbiota DNA extraction method? Do you have any quality control processes?
Line 127, “at t delivery (T0)”, what is t means?
Line 130-131, “37th week (+0 g) to the 42nd week”, what is (+0 g) mean?
Line 378, missing space
Author Response
Pavia, 5 August 2022
Dear Editor, with pleasure, we resubmit the revised version of our study protocol entitled “LIMIT (LIfestyle and Microbiome InTeraction) Early Adiposity Rebound in Children: The Study Protocol” by De Giuseppe R. et al.’s (metabolites-1832247) to be taken into consideration for the publication in “Nutrition during Pregnancy and Offspring Growth and Metabolism” special issue.
The Reviewer’s comments are provided below (in bold the Reviewer’s comments, R, and in plain text our answer, A).
Furthermore, Dr Laura Bertuzzo and Dr Marcello Chieppa will be added to this manuscript as they contributed, together with the other co-authors, in a substantial way to the revision process and will bring added value to the study both from a methodological point of view and as regards the data interpretation.
For this reason, the "Authorship Change Form" format has been completed. We hope the manuscript will be suitable for publication in the present form.
Neither the manuscript nor any significant part of it has been previously published or is under consideration for publication elsewhere; the manuscript has been revised by an English native speaker, read, and approved by all the Authors who have taken due care to ensure the integrity of their work and their scientific reputation. Thank you for your kind attention and consideration.
Your Sincerely,
Rachele De Giuseppe
Dietetics and Clinical Nutrition Laboratory,
Department of Public Health, Experimental and Forensic Medicine,
University of Pavia;
via Bassi 21, 27100 Pavia, Italy.
Email address: rachele.degiuseppe@unipv.it

Reviewer 2 Report
The manuscript entitled „ LIMIT (LIfestyle and Microbiome InTeraction) Early Adiposity 2 Rebound in Children: The Study Protocol” presents the study protocol to be carried out on 272 pairs of Mother-child enrolled at the Neonatal Unit, Fondazione IRCCS Policlinico San Matteo, Pavia (Italy). The protocol Protocol takes into account n many different variables such as time (0, at delivery; T1, one month; T2, six months; T3, 12). months; T4, 24 months; T5, 36 months after birth), anthropometric data, lifestyle, and maternal environmental endocrine disruptors exposure. The protocol has been carefully planned and can serve as a useful tool for collecting not only scientific data but also for predicting the effects of obesity in children in the future. Due to both the size of the study and its assumptions, I will allow myself to suggest a few additional possibilities that the authors may take into account in the study and which can supplement the data set, making it more reliable. From a physiological point of view, the protocol has several weaknesses.
Minor: The Authors indicated that they will take into account maternal environmental endocrine disruptors exposure, however, without examining the biochemical and hormonal profile of blood, it seems impossible to determine whether these changes are due to environmental influences or congenital disorders. It seems that ignoring this issue will greatly limit the "attractiveness" of the proposed research. - I am asking for the Authors' comments on this matter.
Suggestions: Perhaps it would also be worth examining the changes in the hormonal profile in the umbilical cord blood and maternal blood on the day of birth, perhaps it will allow the identification of potential hormonal markers on the day of delivery. Perhaps this will allow the issue of "prenatal programming" to be addressed or eliminated by altering the hormonal profile
Author Response

(The authors gave the same response as above.)

Reviewer 3 Report
This longitudinal comprehensive study protocol is aimed to determine the association between early adiposity rebound (EAR) and early microbiome changes related to maternal and infant lifestyle and environmental variables.
This is an interesting objective, but some important issues need to be addressed or clarified.
Major queries:
- As this is a study protocol and the study results are unknown, it is inaccurate to state (line 108) that ‘The acquired knowledge will allow…’ and ‘will’ should be replaced with ‘may’.
- Growth charts that will be used to classify ‘intrauterine growth retardation’ (line 136) and ‘infants’ weight <10th percentile’ (line 137) should be specified.
- ‘2.1.1. Infant Auxological Parameters’. To accurately evaluate EAR, reliable body mass index (BMI) values are of utmost importance. As length is squared in BMI, any error in length measurement will be magnified while losing the ability to differentiate overestimation from underestimation (Pereira-da-Silva 2018). In this context, if more than one observer is to participate in children anthropometry, the inter-observer coefficient of variation should be previously determined.
- ‘2.1.3. Maternal Anthropometric Parameters’. The Institute of Medicine and National Research Council (2009) have reexamined the pregnancy weight guidelines according to weight gain patterns before, during, and after pregnancy and maternal and child health outcomes. Why were the recommended pre-pregnancy BMI and GWG categories in these guidelines not considered in this protocol to classify maternal overweight/obesity and excessive GWG?
- ‘2.1.9. Infant Dietary Habits’. The method for assessing children diet is very poor. Intake of energy-rich foods and excessive intake of animal protein, particularly protein from cow's milk formula, are reported to be among the main causes of early excessive weight gain, including in Mediterranean European countries (Weber 2014, Pereira-da -Silva 2016, Rolland-Cachera 2016). Therefore, an adequate quantitative dietary assessment of these intakes is necessary to evaluate their influence on early excessive weight gain.
- The timing of adiposity rebound is reported to be an early childhood manifestation of the genetic susceptibility to adult obesity (Couto Alves 2019, Cissé 2021). If this factor will be not assessed it should be acknowledged as a study limitation.
- The Discussion section of a study protocol should refer to the study hypotheses, the strengths and limitations of the methods used, and not focus on theoretical considerations already covered in the Introduction that should not be repeated or should be summarized.
Minor queries:
- In the Abstract (lines 30-31), I suggest replacing ‘at the Neonatal Unit, Fondazione IRCCS Policlinico San Matteo, Pavia (Italy)’ with ‘an Italian neonatal unit’
- I n the manuscript I suggest replacing ‘followups’ and ‘follow-ups’ with ‘time points of assessment’.
- As sick and preterm infants were not included (lines 134-139), please clarify which kind of infants were enrolled from the UOC Neonatology and Neonatal Intensive Care.
- Reference 54 (line 184) should be replaced since it refers to a method for height measurement in neurologically impaired children, not appropriate for participants of this study.
References
- Cissé AH, Lioret S, de Lauzon-Guillain B, Forhan A, Ong KK, Charles MA, Heude B. Association between perinatal factors, genetic susceptibility to obesity and age at adiposity rebound in children of the EDEN mother-child cohort. Int J Obes (Lond). 2021 Aug;45(8):1802-1810. doi: 10.1038/s41366-021-00847-w. Epub 2021 May 13.
- Couto Alves A, De Silva NMG, Karhunen V, et al.; Early Growth Genetics (EGG) Consortium. GWAS on longitudinal growth traits reveals different genetic factors influencing infant, child, and adult BMI. Sci Adv. 2019 Sep 4;5(9):eaaw3095. doi: 10.1126/sciadv.aaw3095.
- Farella I, Miselli F, Campanozzi A, Grosso FM, Laforgia N, Baldassarre ME. Mediterranean diet in developmental age: a narrative review of current evidences and research gaps. Children (Basel). 2022 Jun 16;9(6):906. doi: 10.3390/children9060906. PMID: 35740843; PMCID: PMC9221965.
- Institute of Medicine (US) and National Research Council (US) Committee to Reexamine IOM Pregnancy Weight Guidelines. Weight Gain During Pregnancy: Reexamining the Guidelines. Rasmussen KM, Yaktine AL, editors. Washington (DC): National Academies Press (US); 2009.
- Pereira-da-Silva L, Rêgo C, Pietrobelli A. The diet of preschool children in the Mediterranean countries of the European Union: a systematic review. Int J Environ Res Public Health. 2016 Jun 8;13(6):572. doi: 10.3390/ijerph13060572.
- Pereira-da-Silva L, Virella D. Accurate Direct Measures Are Required to Validate Derived Measures. Neonatology. 2018;113(3):266. doi: 10.1159/000485667.
- Rolland-Cachera MF, Akrout M, Péneau S. Nutrient Intakes in early life and risk of obesity. Int J Environ Res Public Health. 2016 Jun 6;13(6):564. doi: 10.3390/ijerph13060564.
- Weber M, Grote V, Closa-Monasterolo R, Escribano J, et al.; European Childhood Obesity Trial Study Group. Lower protein content in infant formula reduces BMI and obesity risk at school age: follow-up of a randomized trial. Am J Clin Nutr. 2014 May;99(5):1041-51. doi: 10.3945/ajcn.113.064071.
Author Response

(The authors gave the same response as above.)

Round 2
Reviewer 3 Report
The revised manuscript is improved and the questions were clarified.